# Edible Insect Farming in the Context of the EU Regulations and Marketing—An Overview

**DOI:** 10.3390/insects13050446

**Published:** 2022-05-07

**Authors:** Krystyna Żuk-Gołaszewska, Remigiusz Gałęcki, Kazimierz Obremski, Sergiy Smetana, Szczepan Figiel, Janusz Gołaszewski

**Affiliations:** 1Department of Agrotechnology and Agribusiness, Faculty of Agriculture and Forestry, University of Warmia and Mazury in Olsztyn, 10-719 Olsztyn, Poland; kzg@uwm.edu.pl; 2Department of Veterinary Prevention and Feed Hygiene, Faculty of Veterinary Medicine, University of Warmia and Mazury in Olsztyn, 10-719 Olsztyn, Poland; kazimierz.obremski@uwm.edu.pl; 3German Institute of Food Technologies—DIL e.V., 49610 Osnabrück, Germany; s.smetana@dil-ev.de; 4Department of Economics Theory, Faculty of Economic Sciences, University of Warmia and Mazury in Olsztyn, 10-719 Olsztyn, Poland; sfigiel@uwm.edu.pl; 5Center for Bioeconomy and Renewable Energies, Department of Genetics, Plant Breeding and Bioresource Engineering, Faculty of Agriculture and Forestry, University of Warmia and Mazury in Olsztyn, 10-719 Olsztyn, Poland; janusz.golaszewski@uwm.edu.pl

**Keywords:** consumer acceptance, legal regulations, edible insect sector, logistics, marketing, novel food, nutritional value

## Abstract

**Simple Summary:**

Insects have been identified as an alternative in the development of food systems and as a response to the growing demand for protein in the world. Edible insects have been recognized as an important innovation in the food sector. In the past, insects have been consumed in many cultures, and they are presently being introduced to Europe as a novel food and livestock. This article comprehensively reviews the use of edible insects in relation to their breeding, production technology, legal and socio-economic aspects. The role of food safety and legislation in implementing insects as food and feed is discussed. Moreover, the article introduces the breeding of edible insects as a developing and future-oriented business sector. In conclusion, the consumption of insects by humans and animals can significantly contribute to better diversification and security of the global food chain. The low acceptance of insect-based foods, in particular in Western societies, is an important problem that has been identified in this article. Consumer acceptance of insects as a rich source of nutrients is required for the further development of the sector. Consumer education and appropriate marketing strategies are required to promote the growth of the edible insect industry.

**Abstract:**

Insects are increasingly being considered as an attractive source of protein that can cater to the growing demand for food around the world and promote the development of sustainable food systems. Commercial insect farms have been established in various countries, mainly in Asia, but in Europe, edible insects have not yet emerged as a viable alternative to traditional plant- and animal-based sources of protein. In this paper, we present an interdisciplinary overview of the technological aspects of edible insect farming in the context of the EU regulations and marketing. Based on a review of the literature, we have concluded that edible insect farming can be a viable business sector that significantly contributes to the overall sustainability of food systems if the appropriate regulations are introduced and food safety standards are guaranteed. However, the success of the edible insect industry also requires consumer acceptance of entomophagy, which is rather low in Western societies. Therefore, targeted marketing strategies are indispensable to support the implementation of edible insect programs.

## 1. Introduction

Social and demographic changes and the progressive decline in the availability of cultivable land pose a challenge for the food economy. At the same time, the growing demand for various types of foods is becoming an indispensable element of building food security strategies based on sustainable agricultural production and the search for alternative protein sources [1]. The COVID-19 pandemic might also significantly compromise social and economic development, which will affect the functioning of food production systems around the world [2]. From the ecological perspective, it is estimated that by 2050, approximately 150 million people around the world could be at risk of dietary protein deficiency due to rising levels of atmospheric CO_2_ [3]. Numerous research studies have identified significant threats to the stability of the global food chain, including a decrease in the production of dietary protein [1,4,5]. As early as in 1975, Meyer-Rochow [6] suggested that insect farming could offer an alternative solution to the global protein deficit [7,8,9,10]. Insects have been consumed by humans and their primate relatives since time immemorial [11,12,13]. The introduction of insect protein as a rich source of nutrients to modern food chains may eliminate the protein gap. At the same time, edible insects may be revived as an alternative food source and a component of food products, in particular in Western civilizations [14,15,16]. Entomophagy is a new component of the food production system in the European Union. Edible insects have been classified as farmed animals. Previously, bees were the only insects classified as livestock. Edible insects are regarded as novel foods, and their nutritional potential might be utilized in the production of dog, fish, poultry and pig feed, and in human nutrition.

The nutritional value of insects is determined by insect species, production method and the applied feed. The quality and marketability of edible insects is influenced by various factors [17]. Therefore, new knowledge about the nutritional value of feed plays an important role in the development of the insect farming sector. Generally, insects can be fed with a mix of different feedstock, including not-edible primary biomass and agricultural processing waste to minimize the seasonality of insect feed supply [18]. However, the selection of farmed insect species must comply with legal regulations concerning food safety and consumer safety. It should be noted that insect protein is classified as processed animal protein (PAP), which is a source of three primary nutrients: protein, fat and minerals. These nutrients must conform to legal regulations concerning the prevention and eradication of bovine spongiform encephalopathy (BSE). These regulations stipulate that insects may not be directly fed to other farmed animals by default.

Insect harvesting from the natural environment (wild habitats) is not justifiable, and the obtained resources have marginal market value. For this reason, large-scale mass production (farming) of edible insects is regarded as a viable solution. However, farmed insects should meet food safety requirements, which poses considerable technological, social, and economic challenges. This new protein source requires the development of new value chains and solutions concerning production costs, food safety, scalability, and consumer acceptance [1]. The production of edible insects might also be limited by legal regulations and social factors [19]. The knowledge about edible insect production technologies is scant, and producers have limited experience regarding the distribution and sale of edible insects in Europe [20]. There are numerous obstacles and challenges associated with the evolution of food products on the market, which is why the development and performance of the insect farming sector is difficult to predict. At the same time, to the best of the authors’ knowledge, there are no integrative studies analyzing both legal and marketing aspects, future development or potential lock-ins in insect production in Europe.

This paper scrutinizes the recent scientific literature and legal documents to assess the emerging market of edible insects in the context of EU regulations and marketing. The analysis begins by describing the insect industry’s operational, technical, and technological capacities and the environmental aspects of insect farming. The role of legal regulations, marketing and market opportunities is discussed in the following sections. Conclusions concerning the outlook for the insect farming sector are formulated based on an analysis of key events, insect rearing permissions and market penetration trends.

## 2. Technological and Environmental Aspects of Insect Farming

Insect species for large-scale production should be characterized by rapid growth, dietary plasticity, low chitin content, effective bioconversion of feed, high entomoremediation potential, high reproductive performance, a short reproductive cycle and low maintenance requirements [21,22]. The optimal insect species for food production should also be easy to breed, should not require specialist equipment, should feed on a wide range of materials, including wastes from agricultural production and the agri-food industry (plant and animal products), should be resistant to changes in the microclimate of breeding facilities, and should have a high potential for use in human and animal nutrition [21,22,23].

The availability (seasonal, year-round, limited) and form of feed (liquid, dried), rearing conditions (location issues), organizational standards in farming facilities, and the implementation of safety, hygiene and cleaning procedures are often the decisive parameters in insect rearing [24].

The knowledge regarding the safety of biological and chemical residues should be expanded, which poses the greatest challenge in the production of edible insects. Insects such as *H. illucens* may accumulate heavy metals. Purschke et al. did not identify pesticides and mycotoxins in *H. illucens* larvae [25]. However, the end product might be contaminated if insects consume harmful substances directly before harvesting. Moreover, little is known about metabolic pathways in insects, and even if toxic substances are not identified, their metabolites may persist [26]. It should also be noted that insects may be a reservoir of selected substances and pathogens immediately after exposure [10]. Insects may pose a biological threat [10,27,28], and they have even been examined for the ability to transmit COVID-19 infections [10]. Consequently, insect farms must comply with binding regulations, in particular those that set microbiological criteria for food and feed production.

Processing increases consumer acceptance of insects, and processed insects are used in the production of various types of bread, pasta, chips, protein bars and other products in the rapidly growing insect industry [29,30]. Nutrient extraction is an important consideration because high chitin concentration may adversely affect the digestibility of insect-based products [31]. Insects also constitute an excellent source of animal feed because they are an important part of animal diets in nature.

At present, insects are regarded as the poultry feed of the future [32]. Broilers fed with fat extracted from *H. illucens* were characterized by high carcass yield, favorable intestinal morphology and optimal histological parameters of the gastrointestinal tract [33,34]. Research has demonstrated that the Black Soldier Fly (BSF) could be a valuable component of layer and broiler and pig diets [33,34,35].

Protein production in conventional livestock farming and plant cultivation systems requires scarce natural resources (such as land, water, fossil fuels). Continued resource depletion may lead to potentially unfavorable environmental impacts in the future. In this context, insect farming emerges as an environmentally friendly alternative. To minimize their environmental impact, insect farms should rely on energy-efficient heating and lighting systems, and they should be provided with ventilation and air filter installations, sterilization equipment and pest control devices. The environmental benefits of insect farming include effective bioconversion of feed, in particular waste products, as well as lower demand for agricultural land in comparison with conventional animal production [35,36].

Edible insects could play an important role in the bioconversion of agri-food wastes whose global production is estimated at 1.3 billion tons per year [37,38,39]. Financial and logistic considerations play an important role in the selection of insect feed. In the EU, several regulations have been introduced to guarantee the safety of animal feeds, including Commission Regulation (EC) No. 1069/2009, Commission Regulation (EU) No. 142/2011, Commission Regulation (EC) No. 999/2001, Commission Regulation (EC) No. 1137/2014 amending Annex III of Regulation (EC) No. 853/2004, Regulation (EC) No. 767/2009, and Commission Notice (EU) 2018/C 133/02 [40,41,42,43,44,45,46]. These legal acts introduce feed safety standards in the production of edible insects for human and animal consumption. The implemented standards significantly limit the applicability of many conventional food products as insect feed. According to Makkar [38], products such as wheat flour, soymeal and skimmed milk cannot be regarded as sustainable feeds in insect farming because they are a part of the human diet.

Insects effectively convert various types of waste biomass, and they can transform agri-food discards into nutritious products (Table 1). Thus, they could close nutrient cycles, significantly minimize the environmental impacts of food production and reduce production costs [19].

Sustainable insect farming requires dedicated infrastructure and resources, including buildings, equipment and personnel [52,53,54]. Insect production facilities should have access to electricity, water, waste and wastewater handling systems, and a feed logistics network [44]. Insect farms should be established in locations that are free of dust, odor and chemical pollution. They should not be built in the direct vicinity of landfills or waste processing sites. Insect farms should be fenced and protected from insect pests.

Insect production should be effectively managed to maximize yields and profits and to obtain products that meet food safety requirements [55]. Safe food production requires the implementation of dedicated management systems, including good breeding practices, good hygiene practices, and HACCP [56,57]. Food traceability and recall procedures should be put in place as a risk management tool to identify undetected problems and contain food safety problems after the product has reached the food distribution chain [58].

Farm buildings and production facilities have to comply with hygiene and biosecurity standards imposed by the relevant regulations [59]. All premises have to be free of contamination, and they should be easy to clean and disinfect. The design and structure of insect breeding facilities should prevent cross-contamination from other animal production sites. Insect farms should be divided into clean (white) and dirty (black) zones [60]. Buildings should be equipped with air filters and should provide ample working space. All facilities should be monitored for dust contamination, leaks and pests.

Pest control and eradication systems play a very important role in food and feed production [44,61,62]. Such systems should protect insect farms against external sources of pests, and they should prevent farmed insects from escaping. The most common pests in insect farms include other insect species, spiders, birds, rodents and small mammals, which might significantly influence the safety of the end product [63,64]. Disinfection and rodent extermination programs should be implemented to minimize these risks. Production facilities should be maintained in good order and should conform to the applicable standards. Food residues should be promptly eliminated, unnecessary equipment and materials should be removed from production facilities, and organic waste containers should be tightly closed [44,61].

## 3. Role of Legal Regulations

Legal regulations are the key prerequisite for the development of insect farming and effective marketing of insect-based foods. According to Commission Regulation (EU) 2017/893 of 24 May 2017 [14] amending Annexes I and IV to Regulation (EC) No. 999/2001 [42] of the European Parliament and of the Council and Annexes X, XIV and XV to Commission Regulation (EU) No. 142/2011 [41] as regards the provisions on processed animal protein, the following insect species fulfil safety conditions for insect production for feed use: *H. illucens*, *M. domestica*, *T. molitor*, *A. diaperinus*, *G. sigillatus*, *A. domesticus*, and *G. assimilis*. These species were classified as novel foods in the EU based on national risk assessments and the scientific opinion of the European Food Safety Authority (EFSA) of 8 October 2015 [65]. The selected species do not transmit pathogens characteristic of plants, animals or humans; they are not invasive; they do not cause diseases in humans or animals; they do not exert adverse effects on crops; and they are not protected. According to the scientific opinion of EFSA of 13 January 2021 [66], yellow mealworm larvae are safe for consumption. Products classified as novel foods are easier to register. It should also be noted that not only whole insects, but also insect parts and insect-based products (such as insect meal) can be placed on the EU market [67]. Many African countries have developed a solid legal base for harvesting wild-living edible insects [68].

The regulatory framework should address the entire production chain, starting from two different sources of primary production: wild harvest and farmed insects [68]. It is worth noting that similarly to conventional livestock, edible insects should be monitored for food and feed safety [57,69]. Legal regulations addressing edible insect farming and the use of insect-based products were introduced in response to the rapid development of the edible insect industry [67]. Many European countries have not implemented legal regulations concerning edible insects, and they have to abide by the EU laws. Regulatory differences between countries are an important determinant in the geography of launch patterns and in the resulting global assortment of insect products available [70]. Most legal regulations address hygiene standards in feed and food production (Regulation (EC) No. 1069/2009 [40]; Commission Regulation (EU) No. 142/2011 [41]; Commission Implementing Regulation (EU) 2017/2469 [71]). As a result, safety standards can be implemented in the food processing industry, and the range of insect-based products might be expanded [60]. The safety of the introduced products is evaluated by the competent authorities (such as the Veterinary Inspectorate or the Sanitary Inspectorate). The quality of harvested insects, in particular the presence of residues and contaminants, is critical for food safety, and insects should be regularly tested, for example based on Codex Alimentarius recommendations [68]. The detailed schedule of releasing the legal documents for insect breeding in Europe is provided in Table 2.

From the consumers’ point of view, food safety is the most important consideration [72,73,74]. Therefore, insect-based products have to be appropriately certified before they are placed on the market. While insects intended for private consumption are not registered, insects that are placed on the market have to comply with feed and food hygiene standards, good breeding practices, good hygiene practices and good production practices [75,76]. Insects intended for food and feed production cannot be fed materials acquired outside the food chain. Edible insects are classified as livestock; therefore, only plant- and animal-based materials that have been approved for livestock nutrition might be fed to edible insects intended for food and feed production. Commercial feeds for insects have to be purchased from certified manufacturers who apply HACCP principles and European feed laws [75,76] that have been implemented more than 30 years ago (Table 2). Similar to conventional livestock farmers, insect producers are required to store documents confirming feed delivery dates, feed manufacturer and initial feed parameters. Products that do not meet safety standards (recalled feeds, moldy feeds) may not be used in insect feeding. Every batch of insects placed on the food or feed market must conform to microbiological safety standards and the maximum residue limits (MRL) set forth by the relevant regulations [22]. Farmed insects should be periodically monitored for the presence of undesirable chemical substances, including heavy metals, pesticides and mycotoxins [75,76]. Every product batch should be traceable [58].

Insect farmers have to observe biosecurity principles to guarantee that insects meet food safety requirements. Production facilities should be divided into clean and dirty zones to protect insects against pests. Special attention should be paid to the prevention of pathogen transmission to the farm. Edible insects may not be placed on the market if they pose a health risk for humans or animals. Breeders should implement biosecurity measures to protect the farm against insect-specific pathogens. Effective solutions should also be put in place to prevent insects from escaping from the farm.

Insects and insect-based products are classified as Category 3 materials, i.e., by-products derived from animals that do not display any symptoms of disease. The above implies that edible insects might be used as animal feed. Insect protein is classified as PAP, which poses the greatest obstacle to its widespread use. The first regulations limiting the use of PAP in feed production were introduced during the outbreak of transmissible spongiform enteropathies, including BSE. All feeds containing PAP were withdrawn from the market. At present, farmers can use PAP in the production of feed for fish, companion animals and fur-bearing animals. Insect fat and hydrolyzed protein can be administered to all animals. The EU has recently lifted the ban on the use of PAP from one animal species in feeds intended for other animal species [77]. This decision has direct implications for animal breeders because PAP can now be applied in cross-feeding, where the reared species can be fed PAP obtained from other species (excluding ruminants and fur animals) [77]. This regulation is groundbreaking both for future studies and future market perspectives. As a result, insect proteins could be approved for use in pig and poultry diets.

## 4. Crucial Aspects of Edible Insect Marketing

### 4.1. The Product

The global consumption of insects is nearly impossible to establish [87]. At present, more than 2000 insect species are consumed in more than 80 countries [88]. The most widely consumed insects belong to the genera *Coleoptera* (31% of total consumption), *Lepidoptera* (18%), *Hymenoptera* (14%), *Orthoptera* (13%), *Hemiptera* (10%), *Isoptera* (3%), *Odonata* (3%), *Diptera* (2%), and other insect orders that account for around 6% of global insect consumption [89]. The applicability of drone brood from apiaries as a potential food source has been examined in the European Union [90]. There are 1500 edible insect species in Africa [91]. Takeda and Sato [92] found that the seasonal availability of selected insect species was directly correlated with the practice of entomophagy. Some African diets contain insects that are harvested from the natural environment as well as farmed insects. Kinshasa, the capital city of the Democratic Republic of Congo, is a prime example of the above, where an average household consumes around 300 g of caterpillars per week. Around 96 tons of insects are consumed in the Kinshasa region alone [93]. The value of the edible insect market in South Korea is estimated at USD 457 million. The Korean insect market is highly innovative, and edible insects are used not only in food and feed, but also in medicine [94]. In South America, the world’s second largest market of edible insects, entomophagy is practiced in Brazil, Colombia, Venezuela, Ecuador, Peru and Mexico [95]. Numerous attempts have been made to modernize the insect farming industry, but the lack of mass production technologies and considerable market fragmentation constitute the greatest obstacles to the growth of the insect sector [96]. Modern Latin American societies regard entomophagy as something that is practiced by “primitive peoples” [95]. In these countries, insects are widely harvested from the environment, but the insect farming sector has also grown considerably in recent years. Entomophagy is practiced in most countries with a tropical climate, whereas in the Western culture, insects are regarded as a culinary curiosity or an exotic dish [21]. In the EU, 500 tons of insect-based foods were manufactured in 2019. According to estimates, around 1 million tons of edible insects will be produced and processed into 260,000 tons of insect-based foods by 2030. These products will reach 390 million consumers. The value of the edible insect market is expected to reach around EUR 2 billion by 2030 [47,97]. Migratory locust (*Locusta migratoria*), mealworm and lesser mealworm larvae are farmed commercially in the Netherlands [7]. Current insect prices illustrate the size and volume of the edible insect market in rural areas and cities, as well as the value of online sales. In Kenya, 1 kg of termites is valued at EUR 10. In Great Britain and Northern Ireland, 70 g of weaver ants (*Oecophylla smaragdina*) might be purchased online for EUR 7.50. In the Netherlands, the price of 50 g of *T. molitor* and *A. diaperinus* is EUR 4.85, and 35 g of *Locusta migratoria*—around EUR 9.99. In the Democratic Republic of Laos, locust is available already at EUR 8–10 per kilogram [48]. In the Netherlands and Belgium, the sales of beetles of the family Tenebrionidae reached EUR 4,703,400 in 2014 (when they were introduced to the market) and EUR 12,121,200 in 2015 (EUR 4.5/kg of larvae). The sales of *Alphitobius laevigatus* alone reached EUR 993,720 in 2014 and EUR 889,200 in 2015 (EUR 6/kg of larvae). The leading producers of beetle larvae were the Netherlands, followed by Czechia and Hungary which exported their produce to the Netherlands [49]. Edible insects are also widely consumed in Northeast India. Although there is no reliable information about the prices of insects in India, such information is available for other Asian countries, including Laos [98]. Restrictions on wildlife hunting are increasingly often introduced in this part of the world. Therefore, the demand for insects as “wild food” continues to increase, which creates new business opportunities and new sources of income [98]. Local people sell edible insects to stores, snack bars, restaurants, and other outlets in large cities. Surprisingly, crickets are more expensive than meat—10 crickets cost LAK 2500 (LAK, kip, is the Laotian currency: LAK 10,000 = USD 1.06), whereas 1 kg of meat can be purchased for LAK 25,000 [98].

### 4.2. Consumer Acceptance and Willingness to Buy Insect Food Products

Consumer acceptance of insect farming and entomophagy has been researched, but not extensively. As described in the literature, numerous factors influence consumer acceptance of entomophagy [99,100,101,102,103,104]. Firstly, they are related to the product and reflect its price and quality, perceived benefits and risks, perceived naturalness, information about the manufacturing process, tangible benefits for the consumer, confidence in natural foods, quality of substitute products, additional value, the product’s ability to meet consumer needs, and its availability [99,100]. Secondly, consumer acceptance is influenced by social norms and the extent to which consumers trust the claims made by institutions, producers, researchers and product users, and can overcome prior attitudes and opinions concerning insects (i.e., unhygienic), social influence, and feelings of fear, dislike and indifference [22,99,100]. Psychological factors also play an important role, including perceived significance of naturalness, food neophobia, environmental attitudes, availability of information about hazards, cultural factors (taboos), and novelty seeking [50,99,100]. According to Grabowski et al. [68], as regards public health surveillance, insects sold on the market, regardless of if gathered or farmed, should be included in regular monitoring programs. Product labeling should follow national guidelines and should display the name of the insect (ideally including the scientific name), the best-before date and information about allergens [68].

European consumers have rather reserved, but different attitudes towards entomophagy. Verbeke [51] demonstrated that in Western societies, younger males who are weakly attached to meat, are more open to novel foods and are concerned about the environmental impact of their food choices are the most likely early adopters of insects as a novel and more sustainable protein source. Piha et al. [101] found that consumers in Northern Europe generally have a more positive attitude towards insect-based foods than consumers in Central Europe. A study conducted in Czechia demonstrated that 11.8% of the respondents consumed insects regularly, whereas 37.8% had an experience with eating insects. Younger respondents and men had more positive attitudes towards entomophagy, and 77.7% of the surveyed consumers were willing to consume meat products derived from insect-fed livestock [102]. In Poland, insects are consumed sporadically and regarded as an exotic curiosity [21]. According to Zielińska et al. [103], Polish consumers have limited knowledge about entomophagy, and neophobia and low awareness levels are the key barriers to insect consumption. In the surveyed group, 15.51% of the consumers had tried insects, and 60% of these respondents believed insects tasted good or very good [104]. In a Belgian study, 61.9% of the respondents were familiar with entomophagy, and 46.6% had a negative attitude towards insect eating. However, 77.7% of the surveyed subjects were willing to try edible insects, which suggests that Belgian consumers have an interest in novel foods [104]. A survey of Dutch and Australian consumers demonstrated that price and quality were the most important considerations, and that the taste of insect-based products was highly acceptable for consumers [50]. The cited authors suggested that consumers are not yet aware of the benefits of entomophagy [50]. Moruzzo et al. [105] observed that acceptance of entomophagy among Italian consumers was highly correlated with neophobia.

Empirical evidence tested in the marketplace using different categories of edible insects and other novel foods is needed to identify the factors that shape consumer acceptance of edible insects and promote the incorporation of edible insects in Western food cultures [106]. Ghosh et al. [99] stated that certain species of insects are a novel and gradually more and more globally acceptable food item, but attempts to popularize insects as food should bear in mind our finding that even traditionally insect-consuming cultures vary with regard to the choices they make in accepting species as edible and that there are reasons for these differences. Processed insects are inconspicuous food ingredients, and they may increase consumers’ acceptance of novel foods [103]. Zielińska et al. [103] have argued that the future of entomophagy will be largely determined by the sensory attributes of insect-based foods, which is why a wide range of insect-based products should be developed by the food processing industry to provide consumers with an opportunity to sample these foods. In a study by Megido et al. [104], consumers were most willing to accept insects in snacks (37%), main dishes (26%) and desserts (23%), and they were least inclined to accept insect-based salads (7%), soups (6%) and unprocessed insects (1%). However, according to review performed by Dagevos [100], Westerners remain hesitant to include insect eating in their daily diet, and consequently, the evidence suggests that the eating of insects is anything but widespread and common, but rather surrounded by unfamiliarity and reluctance. That is why Mancini et al. [107] suggested that most of the demand for edible insects will be from the feed sector.

### 4.3. Supply Logistics and Product Distribution

The logistic operations associated with raw material supplies play an important part in the production of edible insects (Figure 1 and Figure 2). Supply-related decisions should rely on systemic instruments that meet food safety standards, and logistic operations should be integrated with the producer’s quality assurance system [108]. In business strategies, the quality of the supplied raw materials, followed by the reliability and flexibility of supplies should play a key role in logistics operations to maximize profits and guarantee high-quality end products. This approach focuses on the quality of raw materials that meet the existing standards and regulations. According to Fertsch [109], logistics can be defined as the process of transporting goods and people, as well as the accompanying operations and processes. The main goal of logistic operations is to supply goods and services that: (1) are dedicated to consumers, (2) comprise specific products or services, (3) meet specific quantitative and qualitative requirements, (4) are delivered to a given location at a given time, and (5) are affordable. A knowledge of logistic processes and their structure and parameters is required to identify problems in the production and sale of edible insects, and, consequently, to optimize logistic solutions on the edible insect market. Economic, technical, qualitative, social and environmental parameters should be taken into account in both logistics and marketing operations relating to edible insects.

According to Regulations (EC) No. 852/2004 [110] and 183/2005 [111], special emphasis should be placed on the following factors and processes during logistics operations involving edible insects: water supply; staff hygiene; quality control; audit; thermal processing; food waste; general and specific hygiene requirements in facilities where food is handled, prepared and processed; personnel and training; production; document keeping; regulations concerning wrapping and packaging food products; complaint handling and product recalls; storage and transport; devices and equipment; and equipment standards. These regulations lay down the standards that have to be met to ensure the safety of insect-based foods. Harvested and processed insects are suitable for direct consumption by humans and animals, or they might be used as a substrate for the production of processed foods. Raw materials should be stored in a manner that prevents contamination and does not compromise product quality, in a safe and controlled environment where the stored materials can be adequately preserved. The microclimate in storage facilities should be rigorously monitored. Packaging that is consistent with good hygiene practices also plays an important role in logistics operations involving edible insects. The name of the product, the manufacturer, use-by date and storage conditions should be clearly stated on the packaging. The traceability of food ingredients should meet the requirements of Regulations (EC) 853/2004 [44] and 1169/2011 [83]. At this stage, the product should be tested for compliance with food safety standards. Products should be transported in accordance with the guidelines set forth by Regulations (EC) 852/2004 [110] and 183/2005 [111]. Live insects should be transported in line with animal welfare regulations; the duration of transport should be reduced to a minimum; temperature should be controlled during transport; and the means of transport should be equipped with a ventilation system. Live and processed insects are transported to food and feed producers or directly to consumers and animal breeders.

### 4.4. Market Opportunities and Promotion

Continued population growth and increasing demand for food lead to a worldwide protein gap, which is the key driver behind the development of the edible insect market. According to the calculations based on Henchion et al.’s [1] estimates, in the next 30 years, the global supply of protein will have to increase by around 40% to prevent a supply–demand gap and, consequently, an increase in prices is inevitable. Therefore, the compound annual growth rate (CAGR) of the protein market should reach 1.2% to meet the global demand for protein in 2050.

Edible insects are an attractive food source due the wide variety of insect species with a high nutritional value and appealing taste, clean production process, low production cost, high availability, and an environmentally friendly production technology that contributes to a reduction in GHG emissions. A sound knowledge of insect feeding and their potential to convert low-value organic waste into high-protein foods is needed to increase consumer acceptance of edible insects from production to consumption. Therefore, the key market challenges include: (1) increasing consumer awareness about the potential of edible insects in the context of food safety; (2) selection of insect species and implementation of production technologies based on insects’ developmental biology; (3) stability of protein products; (4) effective marketing strategies, in particular the promotion of insect-based foods in the context of food safety (storage); (5) increased variety of insect-based foods; (6) competitive prices; (7) effective management of food waste; and (8) introduction of legal regulations that facilitate international trade in edible insects [112]. Insect farming offers a much more resource-efficient alternative than the production of poultry, pork and beef. Ghosh et al. [99] suggested that insect farming enterprises need to be tailored to the expectations of the consumers.

Collins et al. [113] have argued that the inclusion of insect foods into the human diet makes not only environmental but also business sense. They underlined the importance of multiple marketing strategies, such as early exposure, education, reducing the visibility of insect parts, celebrity endorsement, or peer-to-peer marketing. Education and well-targeted promotion, especially towards young consumers, appear to be key determinants for building wider social acceptance of entomophagy and facilitating the adoption of insect-based foods. In the commercial sector, insect farms and insect products are financially viable due to increasing consumer demand, and capital from investors has facilitated the emergence of many new insect food companies [114]. These observations indicate that social awareness of entomophagy has to be enhanced to encourage consumers to try insect-based products and to introduce insects to the food processing industry. According to Batat and Peter [106], consumer awareness of insect-based foods is determined mainly by availability and accessibility, food literacy, food taboos, food neophobia, food experience and food ideology. Campaigns informing consumers that edible insects are safe for human consumption and deliver health benefits are needed to promote the practice of entomophagy. Marketing strategies for insect-based products should also be developed. Efforts should be made to increase the public’s knowledge about the insect market and entomophagy as a source of novel foods. Educational campaigns should also target farmers to promote the awareness that edible insects offer an alternative to conventional livestock rearing. Marketing strategies addressing young consumers are particularly desirable because they will increase acceptance of entomophagy in the future.

## 5. Concluding Remarks—The Outlook for Insect Farming Sector

At present, domesticated mammalian species are the key source of protein in the human diet [108], and insect protein may effectively counteract a food crisis in the EU. The research challenges related to industrial insect farming are closely associated with food security and legal regulations concerning the emerging market of insect protein [115]. In the EU, the main threats to food security include rapid population growth, increased demand for food, changing consumption patterns, the food waste problem, food safety, and environmental concerns, including climate change, availability of farmland and drinking water, and loss of biodiversity. In conventional food production systems, these problems are systematically addressed by legal regulations, whereas the knowledge and regulations applicable to commercial insect farming are still in their infancy. Therefore, numerous challenges have to be overcome during the emergence of the edible insect market. These problems are associated with policy making, legislative solutions, standardization and certification of mass-produced edible insects, including [48,116,117,118]:interdisciplinary regulations addressing the production of edible insects that integrate food sciences, agriculture, animal production, conventional medicine, forestry, and socioeconomic and environmental sciences, including the interactions between insect growth and development and production conditions in view of population dynamics and nutrient cycling;genetic modification and insect breeding for specific, desirable traits;controlled production conditions, microbiological concerns in successive stages of insect production, and management of insect diseases;safety of insect-based foods, in particular products derived from insects that are fed organic waste;form, functionality and shelf life of insect-based foods;health benefits for consumers, including the digestibility, toxicity, allergenicity of insect-based foods and their impact on the human biome;environmental concerns, including a life cycle assessment of different food products;socioeconomic concerns, including the profitability of insect farming, consumer acceptance, and attitudes towards edible insects in the catering sector;legal regulations, introduction of universal standards and certification requirements for edible insect farming;long-term impact of insect protein consumption on human and animal health;use of edible insects as therapeutic agents and a resource for the production of new medicinal compounds.

Insect welfare standards are difficult to implement in industrial production due to these knowledge gaps. Ethical concerns in insect farming still remain unclear [119]. Welfare and ethical standards have to be developed for each species of produced insects. Logistics operations, such as insect transport, cannot be effectively controlled unless clear regulations are implemented. At present, edible insects remain outside the scope of veterinary regulations that guarantee the safety of livestock production in the European Union.

Future studies should attempt to determine the optimal conditions for insect breeding and processing insects into foods and substrates with desirable functional traits and acceptable sensory attributes, while maintaining a positive balance between economic performance and environmental sustainability [120]. The mass production of edible insects requires further research focusing on optimal processing methods to achieve the best compromise between profitability, functionality, taste attributes, sustainable development and consumer safety [55].

The correlations between psychological mechanisms and positive attitudes towards insect-based foods have never been explored in the scientific literature [121]. Sociological variables that are closely linked with the consumption of edible insects need to be thoroughly analyzed. Further research should also aim to develop effective strategies for building positive consumer attitudes towards edible insects. Online social support for the consumption of edible insects has been addressed in only a few papers [122]. Insect farms need to be regularly controlled to evaluate the impact of insect production on the economy, environment and local communities. Special attention should be paid to the health of farm employees to identify the potentially adverse effects of insect farming. According to the literature, daily handling of edible insects could contribute to allergies [123,124].

The patenting of methods and systems for insect breeding poses a significant barrier to research and may considerably slow down and impede the development of the edible insect industry. The awareness about insects’ role in agricultural ecosystems [125] and contribution to global food security [55] is generally low. Despite the above, edible insect farming could emerge as a viable business sector that contributes to the sustainability of food systems if rigorous food safety standards are put in place and social acceptance of entomophagy increases.

## Figures and Tables

**Figure 1 insects-13-00446-f001:**
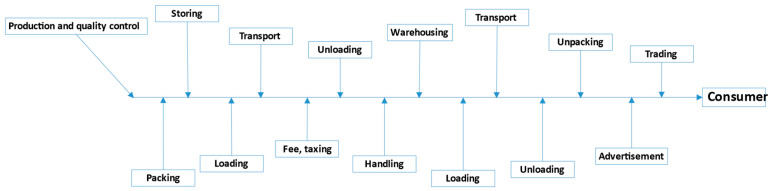
Diagram of basic logistics operations.

**Figure 2 insects-13-00446-f002:**
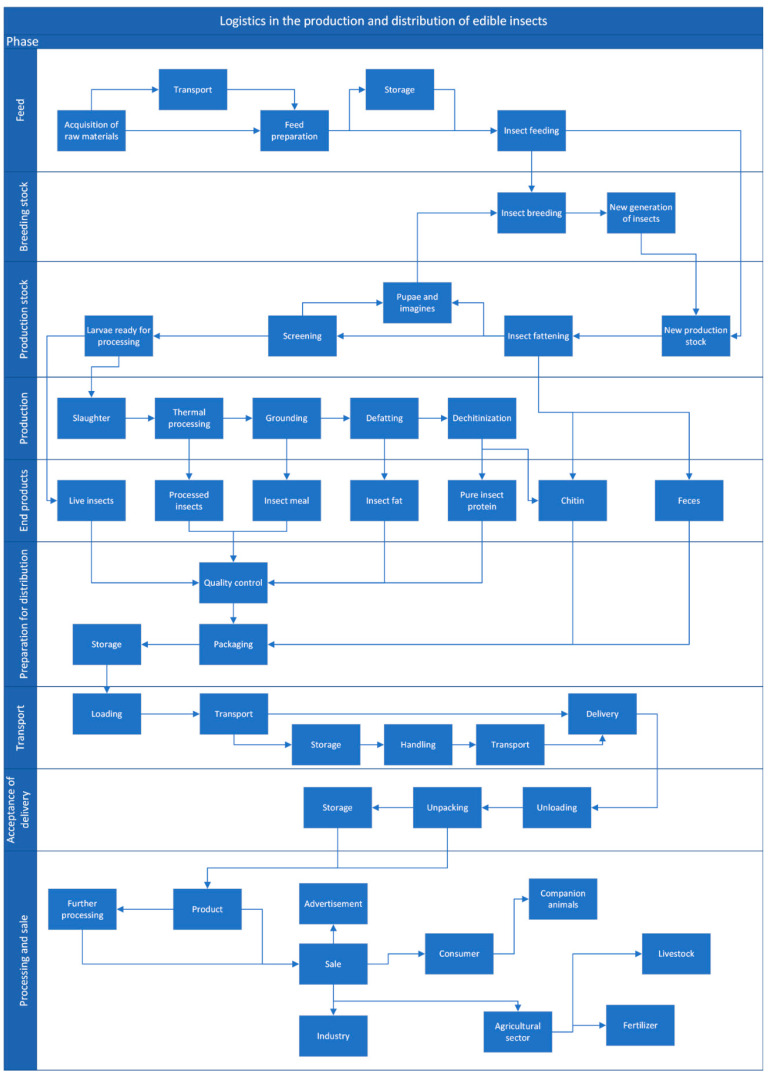
Edible insect farming supply logistics.

**Table 1 insects-13-00446-t001:** Insects capable of converting different waste feedstocks.

Species	Utilization in the Agri-Food Sector		Reference
	Food	Feed	
*Acheta domesticus*	✓	✓	[47]
*Alphitobius diaperinus*	✓	✓	[48]
Cockroaches	✓	✖	[49]
*Hermetia illucens*	✖	✓	[36]
*Musca domestica*	✖	✓	[50]
*Tenebrio molitor*	✓	✓	[51]

**Table 2 insects-13-00446-t002:** Chronologically ordered set of the released legal documents regarding insect production for food and feed.

Name of the Document	Year	Reference
Hazard Analysis and Critical Control Points (HACCP) system and guidelines for its application; practices of conduct in the field of proper animal nutrition; (Codex Alimentarius).	1981	[57]
Regulation (EC) No. 999/2001 of the European Parliament and of the Council of 22 May 2001 laying down rules for the prevention, control and eradication of certain transmissible spongiform encephalopathies.	2001	[42]
Regulation (EC) No. 178/2002 of the European Parliament and of the Council of 28 January 2002 laying down the general principles and requirements of food law, establishing the European Food Safety Authority and laying down procedures in matters of food safety.	2002	[78]
Directive 2002/32/EC of the European Parliament and of the Council of 7 May 2002 on undesirable substances in animal feed—Council statement.	2002	[79]
Regulation (EC) No. 1831/2003 of the European Parliament and of the Council of 22 September 2003 on additives for use in animal nutrition.	2003	[80]
Regulation (EC) No. 882/2004 of the European Parliament and of the Council of 29 April 2004 on official controls performed to ensure the verification of compliance with feed and food law, animal health and animal welfare rules.	2004	[61]
Regulation (EC) No. 853/2004 of the European Parliament and of the Council of 29 April 2004 laying down specific hygiene rules for food of animal origin.	2004	[44]
Regulation (EC) No. 854/2004 of the European Parliament and of the Council of 29 April 2004 laying down specific rules for the organization of official controls on products of animal origin intended for human consumption;	2004	[81]
Commission Regulation (EC) No. 2073/2005 of 15 November 2005 on microbiological criteria for foodstuffs.	2005	[82]
Regulation (EC) No. 1069/2009 of the European Parliament and of the Council of 21 October 2009 laying down health rules as regards animal by-products and derived products not intended for human consumption and repealing Regulation (EC) No. 1774/2002.	2009	[40]
Regulation (EC) No. 767/2009 of the European Parliament and of the Council of 13 July 2009 on the placing on the market and use of feed, amending European Parliament and Council Regulation (EC) No. 1831/2003 and repealing Council Directive 79/373/EEC, Commission Directive 80/511/EEC, Council Directives 82/471/EEC, 83/228/EEC, 93/74/EEC, 93/113/EC and 96/25/EC and Commission Decision 2004/217/EC.	2009	[45]
Regulation (EU) No. 1169/2011 of the European Parliament and of the Council of 25 October 2011 on the provision of food information to consumers, amending Regulations (EC) No. 1924/2006 and (EC) No. 1925/2006 of the European Parliament and of the Council, and repealing Commission Directive 87/250/EEC, Council Directive 90/496/EEC, Commission Directive 1999/10/EC, Directive 2000/13/EC of the European Parliament and of the Council, Commission Directives 2002/67/EC and 2008/5/EC and Commission Regulation (EC) No. 608/2004.	2011	[83]
Commission Regulation (EU) No. 142/2011 of 25 February 2011 implementing Regulation (EC) No. 1069/2009 of the European Parliament and of the Council laying down health rules as regards animal by-products and derived products not intended for human consumption and implementing Council Directive 97/78/EC as regards certain samples and items exempt from veterinary checks at the border under that Directive.	2011	[41]
Regulation (EU) 2015/2283 of the European Parliament and of the Council of 25 November 2015 on novel foods, amending Regulation (EU) No. 1169/2011 of the European Parliament and of the Council and repealing Regulation (EC) No. 258/97 of the European Parliament and of the Council and Commission Regulation (EC) No. 1852/2001.	2015	[67]
EFSA Scientific Opinion “Risk profile related to the production and consumption of insects as food and feed” issued 8 October 2015.	2015	[65]
IPIFF information document “Regulation (EU) 2015/2283 on novel foods- Briefing paper on the provisions relevant to the commercialisation of insect-based products intended for human consumption in the EU.	2015	[84]
Commission Regulation (EU) 2017/893 of 24 May 2017 amending Annexes I and IV to Regulation (EC) No. 999/2001 of the European Parliament and of the Council and Annexes X, XIV and XV to Commission Regulation (EU) No. 142/2011 as regards the provisions on processed animal protein	2017	[85]
Commission Implementing Regulation (EU) 2017/2469 of 20 December 2017 laying down administrative and scientific requirements for applications referred to in Article 10 of Regulation (EU) 2015/2283 of the European Parliament and of the Council on novel foods.	2017	[71]
‘Novel Food’ Report: Opinion on the Risk Profile for House Cricket (Acheta domesticus) by the Swedish University of Agricultural Sciences (EFSA funded report, adopted on 6 July 2018).	2018	[86]
Commission Notice—Guidelines for the feed use of food no longer intended for human consumption C/2018/2035.	2018	[46]
EN ISO 22000: 2018 on food safety management systems.	2018	[69]
Commission Regulation (EU) 2021/1372 of 17 August 2021 amending Annex IV to Regulation (EC) No. 999/2001 of the European Parliament and of the Council as regards the prohibition to feed non-ruminant farmed animals, other than fur animals, with protein derived from animals	2021	[77]

## Data Availability

Not applicable.

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
