# Peer review of "Edible Insect Farming in the Context of the EU Regulations and Marketing—An Overview"

_insects, 2022, doi:10.3390/insects13050446_

Round 1

Reviewer 1 Report

Title: Edible insect farming in the context of the EU legal regulations and marketing – an overview

Authors: Krystyna Żuk-Gołaszewska et al.

This generally well-written manuscript deals with an important  -and until now-  poorly covered aspect, namely EU-regulations and marketing of edible insect species. The article is meant to provide an overview and in several ways it does just that. There are, however, some issues that the authors should address before their manuscript can be deemed acceptable.   

For example, the figure of “148.4 million of people” (Line 54): firstly, delete ‘of’; secondly, why such strange figure 148.4 million? Who can make such an accurate prediction? Why not simply “approx. 150 million people around the world”?

Line 57: The authors ought to insert after “Insect farming…”, “…suggested as early as 1975 by Meyer-Rochow [  ], is emerging as an alternative…”  The missing reference would be Meyer-Rochow, V.B. Can insects help to ease the problem of world food shortage? Search 1975, 6, 261–262.

Line 58: The authors write that insects have been consumed by “native people” and provide a citation. What does ‘native people’ mean? Better write “have been consumed by humans and their primate relatives since time immemorial [ ]. Use perhaps references like:  Bogart, S.L.; Pruetz, J.D. Insectivory of savanna chimpanzees (Pan troglodytes verus) at Fongoli, Senegal. Am. J. Phys. Anthropol2011, 145, 11–20  or better still  McGrew, W.C. The ‘other faunivory’ revisited: Insectivory in human and non-human primates and the evolution of the human diet. J. Human Evol. 2014, 71, 4–11, doi:10.1016/j.jhevol.2013.07.016

Line 64: in connection with the insect species listed, any idea why honey bees and the nowadays also in Europe ‘farmed’ bumble bees are not listed?

Line 68: I think the authors should in addition to dogs and fish mention poultry and hogs (in fact later on Lines 134 and 13 they do!).

Line 69: An excellent review about various factors that influence the quality and marketability of edible insects is this: Meyer-Rochow, V.B.; Gahukar, R.T.; Ghosh, S.; Jung, C.; Chemical Composition, Nutrient Quality and Acceptability of Edible Insects Are Affected by Species, Developmental Stage, Gender, Diet, and Processing Method. Foods 2021, 10, 1036. https://doi.org/10.3390/foods10051036.

Line 80: write “is not justifiable, and the obtained resources have a marginal market value.”  The latter statement is VERY doubtful: even regarding domestic mammals, birds and farmed versus wild fish, the wild products are usually of a higher market value! For honey in East Asia, and even insects, this is true as well.

Line 89: the statement is about Europe, but the reference has noting to do with Europe and refers to Thailand! Either find a suitable reference related to Europe or delete the misleading one.

Line 95: write “This paper scrutinizes the recent scientific literature and …”

Line 98: write “…the insect industry’s operational, technical, and technological capacities and…”

Line 101: use comma after trends “…trends, the study concludes with an outlook  towards the insect farming sector.”

Line 129: you should not forget to mention ‘bread types’ which in Scandinavia can contain 3-5% of cricket protein.

Lines 139 +: On the downside, you should not forget heating and lighting costs; ventilation and air filter installations; sterilization equipment, safety gadgets to keep spiders and vermin away, etc.

Line 194+ : Regarding legal regulations, the authors should read and discuss the paper by Grabowski et al (Grabowski, N.T.; Tchibozo, S.; Abdulmawjood, A.; Acheuk, F.; Guerfali, M.M.; Sayed, W.A.A.; Plötz, M.Edible insects in Africa in terms of food, wildlife resource, and pest management legislation. Foods 2020, 9, 502)

Line 269: It is shocking that this totally insane and inaccurate figure of 2 billion insect-consuming people is still being cited!  Vastly inflated, this grossly erroneous statement is based on, for example, a tiny number of people in Japan from the mountainous provinces of Gifu and Nagano, who very occasionally consume wasps and grasshoppers (perhaps less than 1 million). Does that make all 120 million Japanese regular insect eaters? (I lived in Japan for many years). Or does it justify to include 53 million Koreans as insect consumers, when perhaps a few mostly elderly citizens vey occasionally buy and eat ‘bondaegi’ (silkworms).  Honestly, that figure of 2 billion was ‘invented’ by people with a vested interest to promote insects as food in Europe and it is simply wrong and misleading to cite it.

Lines 275-310: It surprises this reviewer that there seem to be no mention of the insect consumption in North-East India, one of the world’s centres of intensive edible insect consumption.  Maybe there is no information on the price of insects in North-East India, but for some other regions of Asia there is , e.g., Laos:   “….As more and more restrictions are imposed on hunting wild animals, the demand of insects as 'wild food' is increasing and chances to make money from selling them are also increasing. Not surprisingly, 'middle women' set up businesses to buy the insect product and sell it to shops, snack bars, restaurants and other outlets in the big cities (Fig. 19). Perhaps surprisingly, with 10 crickets costing 2,500 Kip (i.e., Laotian currency: 10,000 Kip = 1.06 US$) and 1 kg of meat costing 25,000 Kip, crickets are actually more expensive.”

Section: Consumer acceptance…: The authors are advised to study the comprehensive review by Ghosh et al on “What governs selection and acceptance of edible insects” Chapter 9, pp331-351 in the book Edible insects in sustainable food systems by Halloran A, Flore R, Vantomme P, and Roos N. and include it in their References. That review summarises various reasons for the widespread reluctance of people with western cultural backgrounds to aid insects as food.

Figures 1 and 2: This reviewer wonders whether somewhere in there, the ‘advertising aspect ‘ of edible insects as a part of marketing strategy should not have been mentioned as well?

Regarding the “Concluding Remarks’ this reviewer feels that a little more emphasis could have been given to ‘farmed insects’ as therapeutic agents and a resource of material for new medicinal compounds.

Once adequately revised, this should be an interesting and important article.

Author Response

Proposals of Reviewer 1

  • "This generally well-written manuscript deals with an important  -and until now-  poorly covered aspect, namely EU-regulations and marketing of edible insect species. The article is meant to provide an overview and in several ways it does just that. There are, however, some issues that the authors should address before their manuscript can be deemed acceptable."
    • Thank you very much for the opinion and suggestion of the Reviewer 1. The received review will significantly increase the value of our article. We have made all changes in accordance with the guidelines of Reviewer 1
  • "For example, the figure of “148.4 million of people” (Line 54): firstly, delete ‘of’; secondly, why such strange figure 148.4 million? Who can make such an accurate prediction? Why not simply “approx. 150 million people around the world”?"
    • The mentioned sentence has been corrected as suggested by the Reviewer 1- “From the ecological perspective, it is estimated that by 2050, approx. 150 million people around the world could be at risk of dietary protein deficiency…”
  • "Line 57: The authors ought to insert after “Insect farming…”, “…suggested as early as 1975 by Meyer-Rochow [ ], is emerging as an alternative…”  The missing reference would be Meyer-Rochow, V.B. Can insects help to ease the problem of world food shortage? Search 1975, 6, 261–262."
    • As suggested by the Reviewer, we have added additional information “Insect farming, suggested as early as 1975 by Meyer-Rochow [6] is emerging as an alternative solution to the global protein deficit [7-10]…”.
  • "Line 58: The authors write that insects have been consumed by “native people” and provide a citation. What does ‘native people’ mean? Better write “have been consumed by humans and their primate relatives since time immemorial [ ]. Use perhaps references like: Bogart, S.L.; Pruetz, J.D. Insectivory of savanna chimpanzees (Pan troglodytes verus) at Fongoli, Senegal. Am. J. Phys. Anthropol2011, 145, 11–20  or better still  McGrew, W.C. The ‘other faunivory’ revisited: Insectivory in human and non-human primates and the evolution of the human diet. J. Human Evol. 2014, 71, 4–11, doi:10.1016/j.jhevol.2013.07.016"
    • Many thanks to Reviewer 1 for this suggestion. We corrected this sentence to: “Insects have been consumed by humans and their primate relatives since time immemorial [Bogart et al 2011; McGrew 2014]”.
  • "Line 64: in connection with the insect species listed, any idea why honey bees and the nowadays also in Europe ‘farmed’ bumble bees are not listed?"
    • The concept of using bees (bee brood) as food is familiar in Europe. I personally consumed them during my visit in Finland. Bees have long been recognized as livestock in Europe. Nevertheless, there are several factors that limit bee consumption. The most important is the lack of an economic justification. Bees produce honey, which from the point of view of European consumers is more valuable and more desirable than bee protein. Moreover, collecting bee brood is much more burdensome for bee colonies than obtaining honey. Therefore, this concept does not fit in with the idea of sustainability. The current situation of the bee population, which for various reasons is getting smaller every year, should also be taken into account. Another limiting factor is that bee brood, which is used as food, is usually infected with Varroa destructor. This is partly in contradiction with the current legislation, because in the light of the regulations in force, these animals are sick and therefore cannot be used for consumption. In the manuscript, we added short note that: “Previously, bees were the only insects classified as livestock”.
  • "Line 68: I think the authors should in addition to dogs and fish mention poultry and hogs (in fact later on Lines 134 and 13 they do!)."
    • Although this information is mentioned later, in fact, Reviewer 1 is right that it should also appear here.
  • "Line 69: An excellent review about various factors that influence the quality and marketability of edible insects is this: Meyer-Rochow, V.B.; Gahukar, R.T.; Ghosh, S.; Jung, C.; Chemical Composition, Nutrient Quality and Acceptability of Edible Insects Are Affected by Species, Developmental Stage, Gender, Diet, and Processing Method. Foods 2021, 10, 1036. https://doi.org/10.3390/foods10051036."
    • As requested by Reviewer 1, the citation has been added to the text.
  • "Line 80: write “is not justifiable, and the obtained resources have a marginal market value.” The latter statement is VERY doubtful: even regarding domestic mammals, birds and farmed versus wild fish, the wild products are usually of a higher market value! For honey in East Asia, and even insects, this is true as well."
    • As suggested by Reviewer 1, we have added that sentence. Indeed now the statement is clearer.
  • "Line 89: the statement is about Europe, but the reference has noting to do with Europe and refers to Thailand! Either find a suitable reference related to Europe or delete the misleading one."
    • Thank you very much for showing this citation error. Indeed, the cited article was not relevant to Europe, therefore it has been changed to: Stull, V., & Patz, J. (2020). Research and policy priorities for edible insects. Sustainability Science15(2), 633-645.
  • "Line 95: write “This paper scrutinizes the recent scientific literature and …”"

and

  • "Line 98: write “…the insect industry’s operational, technical, and technological capacities and…”"

and

  • "Line 101: use comma after trends “…trends, the study concludes with an outlook towards the insect farming sector.”"
    • As suggested by Reviewer 1, we have corrected this fragment of the text: “This paper scrutinizes the recent scientific literature and legal documents to assess the emerging market of edible insects in the context of the EU regulations and marketing. The analysis begins by describing the insect industry’s operational, technical, and technological capacities and the environmental aspects of insect farming. The role of legal regulations, marketing and market opportunities is discussed in the following sections. Conclusions concerning the outlook for the insect farming sector are formulated based on an analysis of key events, insect rearing permissions and market penetration trends.”
  • "Line 129: you should not forget to mention ‘bread types’ which in Scandinavia can contain 3-5% of cricket protein."
    • As Reviewer 1 pointed out correctly, we added information about the bread. Thank you for your suggestions.
  • "Lines 139 +: On the downside, you should not forget heating and lighting costs; ventilation and air filter installations; sterilization equipment, safety gadgets to keep spiders and vermin away, etc."
    • We have included the suggestion mentioned by Reviewer 1 in our manuscript: “In this context, insect farming emerges as an environmentally friendly alternative. With regard to the environment, producers should also keep in mind heating and lighting costs; ventilation and air filter installations; sterilization equipment, safety gadgets to keep pests away, etc.”.
  • "Line 194+ : Regarding legal regulations, the authors should read and discuss the paper by Grabowski et al (Grabowski, N.T.; Tchibozo, S.; Abdulmawjood, A.; Acheuk, F.; Guerfali, M.M.; Sayed, W.A.A.; Plötz, M.Edible insects in Africa in terms of food, wildlife resource, and pest management legislation. Foods 2020, 9, 502)"
    • As suggested by Reviewer 1, we have added this article to our manuscript. We cite him in several places: “Regarding gathering from the wild, many African countries already have a solid legal base that could be applied to insects.”; “Regulatory framework should address the entire production chain, starting from two different points of primary production, i.e., wildlife resource and farmed insects”; “For food safety, the quality of the gathered animals is critical, particularly in terms of residues and contaminants, and harvested animals should be tested on a regular base, e.g., using Codex Alimentarius recommendations”; “Grabowski et al stated, that In terms of public health surveillance, insects sold publicly, regardless if gathered or farmed, should be included in regular monitoring programs. Appropriate labeling follows national guidelines and should, by any chance, show the name of the insect (ideally including the scientific name), a best-before date and the information about allergens”
  • "Line 269: It is shocking that this totally insane and inaccurate figure of 2 billion insect-consuming people is still being cited! Vastly inflated, this grossly erroneous statement is based on, for example, a tiny number of people in Japan from the mountainous provinces of Gifu and Nagano, who very occasionally consume wasps and grasshoppers (perhaps less than 1 million). Does that make all 120 million Japanese regular insect eaters? (I lived in Japan for many years). Or does it justify to include 53 million Koreans as insect consumers, when perhaps a few mostly elderly citizens vey occasionally buy and eat ‘bondaegi’ (silkworms).  Honestly, that figure of 2 billion was ‘invented’ by people with a vested interest to promote insects as food in Europe and it is simply wrong and misleading to cite it."
    • I admit that we used the mentioned information indiscriminately. Thank you very much for the reliable explanation of the incorrect assumption. We have read an article by van Huis, A. Halloran, J. Van Itterbeeck, H. Klunder, and P. Vantomme. How many people on our planet eat insects: 2 billion? Journal of Insects as Food and Feed 2022 8: 1, 1-4. https://doi.org/10.3920/JIFF2021.x010. “It was published after we wrote the article. Based on the Reviewer's 1 request and the aforementioned article, we corrected the information: “It is almost impossible to calculate an accurate figure of how many people eat insects globally [van Huis, A. et al. 2022]. At present, more than 2 thousand insect species are consumed in more than 80 countries.”.
  • "Lines 275-310: It surprises this reviewer that there seem to be no mention of the insect consumption in North-East India, one of the world’s centres of intensive edible insect consumption. Maybe there is no information on the price of insects in North-East India, but for some other regions of Asia there is , e.g., Laos:   “….As more and more restrictions are imposed on hunting wild animals, the demand of insects as 'wild food' is increasing and chances to make money from selling them are also increasing. Not surprisingly, 'middle women' set up businesses to buy the insect product and sell it to shops, snack bars, restaurants and other outlets in the big cities (Fig. 19). Perhaps surprisingly, with 10 crickets costing 2,500 Kip (i.e., Laotian currency: 10,000 Kip = 1.06 US$) and 1 kg of meat costing 25,000 Kip, crickets are actually more expensive.”"
    • Thank you for your very valuable comment. Reviewer 1 is right that the mentioned information is important. For this reason, we have included them in the text: “It should also be mentioned that Northeast India is one of the world's centers of intensive consumption of edible insects. Although there is no reliable information about the prices of insects in this country, prices for this region of the world, e.g. from Laos can be found. In this region, more and more restrictions are placed on hunting wild animals. Therefore, the demand for insects as "wild food" is increasing, as well as the chances of earning money by selling them. Local people sell edible insects to stores, snack bars, restaurants, and other outlets in large cities. Perhaps surprisingly, with 10 crickets costing 2500 kip (ie Laotian currency: 10,000 kip = $ 1.06) and 1 kg of meat for 25,000 kip, crickets are actually more expensive.”.
  • "Section: Consumer acceptance…: The authors are advised to study the comprehensive review by Ghosh et al on “What governs selection and acceptance of edible insects” Chapter 9, pp331-351 in the book Edible insects in sustainable food systems by Halloran A, Flore R, Vantomme P, and Roos N. and include it in their References. That review summarises various reasons for the widespread reluctance of people with western cultural backgrounds to aid insects as food."
    • We analyzed the publication in the context of our description and can conclude that our results are largely consistent with the ones described in the mentioned chapter. Based on the suggestion of Reviewer 1, we have added the information from this article along with the citation
  • "Figures 1 and 2: This reviewer wonders whether somewhere in there, the ‘advertising aspect ‘ of edible insects as a part of marketing strategy should not have been mentioned as well?"
    • Based on the suggestion of Reviewer 1, we have added the "advertisement" point to Figure 1 and 2 at the end of logistic routes.
  • "Regarding the “Concluding Remarks’ this reviewer feels that a little more emphasis could have been given to ‘farmed insects’ as therapeutic agents and a resource of material for new medicinal compounds."
    • Due to the remark of Reviewer 3 regarding the length of this chapter, we did not discuss this topic in greater detail. However, having in mind the suggestion, we added a point: “Utilization of edible insects as therapeutic agents and a resource of material for new medicinal compounds”.

Reviewer 2 Report

I've read with great interest the paper and it provides a clear and complete overview regarding edible insects from multiple perspectives. Overall, the paper is well written, readable and of interest. I have just one comment regarding the recent Commission Regulation (EU) 2021/1372, which has been cited in table but it has not been deeply discussed in the text (besides lines 253-260). I would ask the author to spend some thought regards this regulation which, in my opinion, is groundbreaking both for future studies and future market perspectives.

Author Response

Proposals of Reviewer 2

  • “I've read with great interest the paper and it provides a clear and complete overview regarding edible insects from multiple perspectives. Overall, the paper is well written, readable and of interest. I have just one comment regarding the recent Commission Regulation (EU) 2021/1372, which has been cited in table but it has not been deeply discussed in the text (besides lines 253-260). I would ask the author to spend some thought regards this regulation which, in my opinion, is groundbreaking both for future studies and future market perspectives.”
    • Thank you very much for the opinion and suggestion of the Reviewer 2. The received review will significantly increase the value of our article. We have made all changes in accordance with the guidelines of Reviewer 2.
    • The Commission Regulation (EU) 2021/1372 mentioned by Reviewer 2 was better marked in the text (we added citations to the text that were missing for this provision). We have also added some information illustrating the role of this law. This entire paragraph relates to PAP issues and shows the possibilities that this provision has opened: “Insects and insect-based products are classified as Category 3 materials, i.e., by-products derived from animals that do not display any symptoms of disease. The above implies that edible insects might be used as animal feed. Insect protein is classified as PAP, which poses the greatest obstacle to its widespread use. The first regulations limiting the use of PAP in feed production were introduced during the outbreak of transmissible spongiform enteropathies, including BSE. All feeds containing PAP were withdrawn from the market. At present, farmers can use PAP in the production of feed for fish, companion animals and fur-bearing animals. Insect fat and hydrolyzed protein can be administered to all animals. The EU has recently lifted the ban on the use of PAP from one animal species in feeds intended for other animal species [72]. This decision has direct implications for animal breeders because PAP can now be applied in cross-feeding, where the reared species can be fed PAP obtained from other species (excluding ruminants and fur animals) [72]. This regulation is groundbreaking both for future studies and future market perspectives. As a result, insect proteins could be approved for use in pig and poultry diets”.

Reviewer 3 Report

Manuscript: Edible insect farming in the context of the EU legal regulations and marketing – an overview

Dear Authors,

Thank you for the opportunity to review this manuscript which describes both technological, farming, safety, and market aspects of edible insects industry in the context of the EU legal regulations. The review is very comprehensive and the authors clearly pointed out the gaps in the current state of the art. However, some of the statements could be clarified and the overall discussion could be supported through further review of the literature.

First, a comment about the title: why “legal regulations”? is not enough to use “regulation”? Shorter is better and perhaps clearer.

Second, I would recommend not to mention keywords which are already present in the title (edible insects, legal regulations).

Please I recommend the authors to be more specific. Avoid implications like “need of society education”.

Ln58: what does it mean “Insects have been consumed by native people”? Where? The authors need to report regions.

What does it mean “they are a part of the natural food chains.”? Is there a not natural food chain? If so, please the differences.

In general, try to be more critical and objective. Try to avoid strong statement such as “The introduction of insect … can eliminate the protein gap”, “Pesticides and mycotoxins were not detected in insects”, unless very well explained and justified by several studies and not only one. Especially if then it is said “Insufficient knowledge regarding the safety of biological”. Thus, it would be better in some cases taking a stance more cautious.

It is difficult to state whether “no integrative studies analyzing both legal and marketing domains of insect production in Europe” is true or not. However, to my knowledge, there are several articles which try to cover these topics (perhaps with a less focus). Perhaps the authors would like to consider them to integrate the discussion:

  • Lähteenmäki-Uutela, L. Hénault-Ethier, S.B. Marimuthu, S. Talibov, R.N. Allen, V. Nemane, G.W. Vandenberg, and D. Józefiak. The impact of the insect regulatory system on the insect marketing system. Journal of Insects as Food and Feed 2018 4:3, 187-198
  • van Huis, B.A. Rumpold, H.J. van der Fels-Klerx, and J.K. Tomberlin. Advancing edible insects as food and feed in a circular economy. Journal of Insects as Food and Feed 2021 7:5, 935-948
  • Mancini S, Sogari G, Espinosa Diaz S, Menozzi D, Paci G, Moruzzo R. Exploring the Future of Edible Insects in Europe. Foods. 2022; 11(3):455. https://doi.org/10.3390/foods11030455
  • Pippinato, L. Gasco, G. Di Vita, and T. Mancuso. Current scenario in the European edible-insect industry: a preliminary study. Journal of Insects as Food and Feed 2020 6:4, 371-381
  • Payne C., Caparros Megido R., Dobermann D., Frédéric F., Shockley M., Sogari G. (2019) Insects as Food in the Global North – The Evolution of the Entomophagy Movement. In: Sogari G., Mora C., Menozzi D. (eds) Edible Insects in the Food Sector. Springer, Cham DOI: https://doi.org/10.1007/978-3-030-22522-3_2

Pay attention to repetitions, which is the risk in this kind of review. For example ln 60 and ln200.

Pay attention also to mistakes in terms of regulations and dates. For example, it was not 13 January 2020 but 2021. I believe that the previous mentioned suggested literature could help with that (see Mancini et al.).

I’m not sure whether the title of Table 2 “Chronologically ordered set of the released legal documents regarding insect production” is appropriate. Most of the regulations are general and only the last ones are specific to insects (considering that until a few years ago they were not even taken into consideration), which are the ones most important for the readers.

Ln269: Recently the fact that “two billion people around the globe include insects on their diet” has been retracted by van Huis himself. Please consider reading his recent paper:

  • van Huis, A. Halloran, J. Van Itterbeeck, H. Klunder, and P. Vantomme. How many people on our planet eat insects: 2 billion? Journal of Insects as Food and Feed 2022 8:1, 1-4. https://doi.org/10.3920/JIFF2021.x010

Ln 314: “As described in the literature factors influencing consumer acceptance in case of entomophagy are numerous”. However, the authors do not report any referenced literature.

The only citations reported are a little bit outdated from some years ago (e.g., Lensvelt et al. (2014), which is not even in Europe). I would strongly recommend the authors take a look at this recent review:

  • Dagevos (2021). A literature review of consumer research on edible insects: recent evidence and new vistas from 2019 studies

Concluding remarks are very long and in several parts repetitive. the conclusions include elements that go beyond the very content of the previous subsections. In addition, it does not seem EU-focused but more global.

Moreover, in ln 510 “Very few studies have investigated the role of consumer attitudes towards entomophagy”. This is not true. Just consider all the studies included in the abovementioned review by Dagevos.

Minor comments:

The topic of the review is relevant, but sometimes the quality of the writing prevents it from being communicated well to the reader. For example: “to support the adoption of insect food public education and properly designed and targeted marketing strategies are indispensable.”

There are several writing mistakes; be sure to check both language, wording, spelling, and punctuation. For example in the Simple Summary: the article introduces…

Ln 340: mistake in parenthesis

I would recommend some additional review of the manuscript for these and related errors prior to publication.

Moreover, to facilitate the readers and have a better flow, the authors should try to reduce the length of some sentences. For example, at the beginning of the section 2. Technological and environmental aspects of insect farming, there are several sentences very long; please consider splitting the concepts and link them with academic transition words.

Improve the quality of Figure 2 with larger font size.

Finally, report always the original source. For example, in ln 414 the authors should refer to the FAO source

Author Response

Proposals of Reviewer 3

  • "“Thank you for the opportunity to review this manuscript which describes both technological, farming, safety, and market aspects of edible insects industry in the context of the EU legal regulations. The review is very comprehensive and the authors clearly pointed out the gaps in the current state of the art. However, some of the statements could be clarified and the overall discussion could be supported through further review of the literature."
    • Thank you very much for the opinion and suggestion of the Reviewer 3. The received review will significantly increase the value of our article. We have made all changes in accordance with the guidelines of Reviewer 3.
  • "First, a comment about the title: why “legal regulations”? is not enough to use “regulation”? Shorter is better and perhaps clearer."
    • Based on the opinion of Reviewer 3, we changed the title of the article to: “Edible insect farming in the context of the EU regulations and marketing – an overview”.
  • "Second, I would recommend not to mention keywords which are already present in the title (edible insects, legal regulations)."
    • We changed "edible insects" to "novel food" as suggested by Reviewer 3. Due to the fact that we removed the word "legal" in the title, we decided to leave it in the keywords.
  • "Please I recommend the authors to be more specific. Avoid implications like “need of society education”."
    • We tried to clear the article from general statements based on request of Reviewer 3.
  • "Ln58: what does it mean “Insects have been consumed by native people”? Where? The authors need to report regions."
    • Due to the suggestions of Reviewer 1, this sentence has been changed to: “Insects have have been consumed by humans and their primate relatives since time immemorial [Bogart et al 2011; McGrew 2014]”.
    • Bogart, S.L.; Pruetz, J.D. Insectivory of savanna chimpanzees (Pan troglodytes verus) at Fongoli, Senegal. Am. J. Phys. Anthropol 2011, 145, 11–20
    • McGrew, W.C. The ‘other faunivory’ revisited: Insectivory in human and non-human primates and the evolution of the human diet. J. Human Evol. 2014, 71, 4–11, doi:10.1016/j.jhevol.2013.07.016
  • "What does it mean “they are a part of the natural food chains.”? Is there a not natural food chain? If so, please the differences."
    • Based on the opinion of the Reviewer 3, this fragment has been removed.
  • "In general, try to be more critical and objective. Try to avoid strong statement such as “The introduction of insect … can eliminate the protein gap”, “Pesticides and mycotoxins were not detected in insects”, unless very well explained and justified by several studies and not only one. Especially if then it is said “Insufficient knowledge regarding the safety of biological”. Thus, it would be better in some cases taking a stance more cautious."
    • Strong sentences have been removed from the article based on the suggestion of the Reviewer 3. In the process of revising the article, we tried to be more critical.
  • "It is difficult to state whether “no integrative studies analyzing both legal and marketing domains of insect production in Europe” is true or not. However, to my knowledge, there are several articles which try to cover these topics (perhaps with a less focus). Perhaps the authors would like to consider them to integrate the discussion:"

Lähteenmäki-Uutela, L. Hénault-Ethier, S.B. Marimuthu, S. Talibov, R.N. Allen, V. Nemane, G.W. Vandenberg, and D. Józefiak. The impact of the insect regulatory system on the insect marketing system. Journal of Insects as Food and Feed 2018 4:3, 187-198

van Huis, B.A. Rumpold, H.J. van der Fels-Klerx, and J.K. Tomberlin. Advancing edible insects as food and feed in a circular economy. Journal of Insects as Food and Feed 2021 7:5, 935-948

Mancini S, Sogari G, Espinosa Diaz S, Menozzi D, Paci G, Moruzzo R. Exploring the Future of Edible Insects in Europe. Foods. 2022; 11(3):455. https://doi.org/10.3390/foods11030455

Pippinato, L. Gasco, G. Di Vita, and T. Mancuso. Current scenario in the European edible-insect industry: a preliminary study. Journal of Insects as Food and Feed 2020 6:4, 371-381

Payne C., Caparros Megido R., Dobermann D., Frédéric F., Shockley M., Sogari G. (2019) Insects as Food in the Global North – The Evolution of the Entomophagy Movement. In: Sogari G., Mora C., Menozzi D. (eds) Edible Insects in the Food Sector. Springer, Cham DOI: https://doi.org/10.1007/978-3-030-22522-3_2

    • Based on the opinion of Reviewer 3, we additionally cited 4 out of 5 suggested articles
  • "Pay attention to repetitions, which is the risk in this kind of review. For example ln 60 and ln200."
    • In our opinion, we have removed all unnecessary repetitions from the text.
  • "Pay attention also to mistakes in terms of regulations and dates. For example, it was not 13 January 2020 but 2021. I believe that the previous mentioned suggested literature could help with that (see Mancini et al.)."
    • Thank you very much to Reviewer 3 for discovering the wrong date. This error occurred while editing the text. We applied appropriate corrections and checked the article for similar mistakes.
  • "I’m not sure whether the title of Table 2 “Chronologically ordered set of the released legal documents regarding insect production” is appropriate. Most of the regulations are general and only the last ones are specific to insects (considering that until a few years ago they were not even taken into consideration), which are the ones most important for the readers."
    • Due to the doubts of Reviewer 3, we changed the title of the table to: Chronologically ordered set of the released legal documents regarding insect production for food and feed.
      We would like to assure Reviewer 3 that all legal documents cited in the table are relevant to the breeding of insects. At the outset, it should be noted that edible insects were included in the livestock category. So the legal provisions on maintenance, circulation and even feed have to be taken into account. We agree that not all laws apply directly to insects, but rather to keeping livestock (and therefore edible insects). Insects must also comply with all veterinary regulations. Therefore, although not all laws apply to insect breeding, all laws apply to the use of insects as food and feed. Therefore, even general regulations such as the Codex Allimentarius or ISO must be included in the review. We know from our own experience, that it is not so obvious for all recipients.
  • "Ln269: Recently the fact that “two billion people around the globe include insects on their diet” has been retracted by van Huis himself. Please consider reading his recent paper: van Huis, A. Halloran, J. Van Itterbeeck, H. Klunder, and P. Vantomme. How many people on our planet eat insects: 2 billion? Journal of Insects as Food and Feed 2022 8:1, 1-4. https://doi.org/10.3920/JIFF2021.x010"
    • I admit that we used the mentioned information indiscriminately. Thank you very much for the reliable explanation of the incorrect assumption. We have read an article by van Huis, A. et al. 2022. It was published after we wrote the article. Based on the Reviewer's 3 request and the aforementioned article, we corrected the information: It is almost impossible to calculate an accurate figure of how many people eat insects globally [van Huis, A. et al. 2022]. At present, more than 2 thousand insect species are consumed in more than 80 countries.”.
  • "Ln 314: “As described in the literature factors influencing consumer acceptance in case of entomophagy are numerous”. However, the authors do not report any referenced literature."
    • We added proper citation in this paragraph, as Reviewer 3 requested
  • "The only citations reported are a little bit outdated from some years ago (e.g., Lensvelt et al. (2014), which is not even in Europe). I would strongly recommend the authors take a look at this recent review:"
  • Dagevos (2021). A literature review of consumer research on edible insects: recent evidence and new vistas from 2019 studies
    • The article presented by Lensvelt et al. applies to both Australia and the Netherlands. As Reviewer 3 suggested we added additional citations including Dagevos 2021. We analyzed the publication in the context of our description and can conclude that our results are largely consistent with the ones described in the mentioned chapter.
  • "Concluding remarks are very long and in several parts repetitive. the conclusions include elements that go beyond the very content of the previous subsections. In addition, it does not seem EU-focused but more global."
    • As suggested by Reviewer 3, the Final remarks chapter has been shortened and the repeated information has been removed.
  • "Moreover, in ln 510 “Very few studies have investigated the role of consumer attitudes towards entomophagy”. This is not true. Just consider all the studies included in the abovementioned review by Dagevos."
    • Due to the expertise of Reviewer 3, this sentence has been removed.
  • Minor comments:
  • "The topic of the review is relevant, but sometimes the quality of the writing prevents it from being communicated well to the reader. For example: “to support the adoption of insect food public education and properly designed and targeted marketing strategies are indispensable.”"

and

  • "There are several writing mistakes; be sure to check both language, wording, spelling, and punctuation. For example in the Simple Summary: the article introduces…"

and

  • "Ln 340: mistake in parenthesis"

and

  • "Moreover, to facilitate the readers and have a better flow, the authors should try to reduce the length of some sentences. For example, at the beginning of the section 2. Technological and environmental aspects of insect farming, there are several sentences very long; please consider splitting the concepts and link them with academic transition words."
    • We would like to assure Reviewer 3 that the article has undergone a thorough proofreading by an additional native speaker. Linguistic errors (language, wording, spelling, and punctuation) were removed, readability has been improved and too long sentences were separated  during proofreading. As recommended by the Reviewer, the manuscript has been edited to eliminate grammatical errors and stylistic inconsistencies, and to improve overall readability and clarity of presentation. It has also been spell-checked and grammar-checked by a native English speaker prior to re-submission.
  • "Improve the quality of Figure 2 with larger font size."
    • The font size has been improved.
  • "Finally, report always the original source. For example, in ln 414 the authors should refer to the FAO source"
    • We changed the sentence to “According to the calculations based on Henchion et al. [1] estimates, in the next 30 years, the global supply of protein will have to increase by around 40% to prevent a sup-ply-demand gap and, consequently, an increase in prices.”

Round 2

Reviewer 1 Report

Excellent piece of work, congratulations.  The interesting information on Laos and costs of edible insects in that country is actually quoted from page 20 of: "More feared than revered: Insects and their impact on human societies (with some specific data on the importance of entomophagy in a Laotian Setting)" by Meyer-Rochow, V.B., Nonaka K, Boulidam, S. In: Entomologie Heute 2008, Vol. 20: pp. 3-25.

Author Response

Dear Reviewer 1,

We sincerely apologize for omitting this citation. As noted by the Reviewer, we have added this article to our manuscript.

Reviewer 3 Report

After checking the manuscript "Edible insect farming in the context of the EU regulations and marketing – an overview" once it was amended by the authors, my suggestion to the editor is to accept it. 

I only noticed that the title in figure2 needs to be edited (reduced size).

Author Response

Dear Reviewer 3

Thank you very much for taking the time to review our article. In line with the comment from Reviewer 3, we updated Figure 2.

This manuscript is a resubmission of an earlier submission. The following is a list of the peer review reports and author responses from that submission.

Round 1

Reviewer 1 Report

The present paper has invested the Edible insect farming in the context of the EU legal regulations 2 and marketing – an overview. The study is interesting but does not fit into the scope of the journal because it is related to animal nutrition or animal sciences journals and also they used self-citation. Such as 1. Doi, H.; Gałęcki, R.; Mulia, R.N. The merits of entomophagy in the post COVID-19 world. Trends Food Sci. Technol. 2021, 110, 669 849-854. 2. Gałęcki, R.; Zielonka, Ł.; Zasępa¸ M.; Gołębiowska, J.; Bakuła, T.; Potential Utilization of Edible Insects as an Alternative Source 694 of Protein in Animal Diets in Poland. Front. Sustain. Food Syst. 2021, 5, 675796. 69. 3. Bakuła, T.; Gałęcki, R. Strategia wykorzystania alternatywnych źródeł białka w żywieniu zwierząt oraz możliwości rozwoju 696 jego produkcji na terytorium Rzeczpospolitej Polski. 2021, ISBN 978-83-961897-1-4 ERZET, Olsztyn. 4. Gałęcki, R.;;Sokół, R. A parasitological evaluation of edible insects and their role in the transmission of parasitic diseases to 744 humans and animals. PloS One 2019, 14(7), e0219303. 5. Pleissner, D.; Smetana, S. Estimation of the economy of heterotrophic microalgae-and insect-based food waste utilization pro-787 cesses. Waste Manage. 2020, 102, 198-203. Overall the study has been not well performed. The English language needs to be corrected thoroughly. The title is not well write abstract is not well performed Discussion is not well compared to others scientist papers. Rewrite the introduction, Materials and Methods, Results, Discussion, Conclusion.

Reviewer 2 Report

The paper is interesting for the legal rules and marketing parts but should be reduced for previous parts. Indeed, sections 2 "Selection of insect species for farming", 3 "Nutritional value and uses" and 4 "Technological and environmental aspects of insect farming" do not bring significant new informations. Several other paper already presented synthesis for theses topics.

Authors are asked to adapt the content of the paper more strictly in relation with the title : Edible insects in the context of the EU legal regulations and marketing.  

Reviewer 3 Report

a really comprehensive overview of the status of entomophagy and of recomendations concerning entomophagy

just some few mistakes in spelling.